# Myostatin (MSTN) Gene Indel Variation and Its Associations with Body Traits in Shaanbei White Cashmere Goat

**DOI:** 10.3390/ani10010168

**Published:** 2020-01-19

**Authors:** Yi Bi, Bo Feng, Zhen Wang, Haijing Zhu, Lei Qu, Xianyong Lan, Chuanying Pan, Xiaoyue Song

**Affiliations:** 1Key Laboratory of Animal Genetics, Breeding and Reproduction of Shaanxi Province, College of Animal Science and Technology, Northwest A&F University, Yangling 712100, China; biyi0312@163.com (Y.B.); Fengbobo927@163.com (B.F.); wangzhenid@126.com (Z.W.); lanxianyong79@126.com (X.L.); 2Shaanxi Provincial Engineering and Technology Research Center of Cashmere Goats, Yulin University, Yulin 719000, China; haijingzhu@yulinu.edu.cn (H.Z.); ylqulei@126.com (L.Q.); 3Life Science Research Center, Yulin University, Yulin 719000, China

**Keywords:** goat, *MSTN* gene, 5 bp indel, growth, correlation

## Abstract

**Simple Summary:**

As a major gene to regulate the muscle mass of animals, Myostatin (*MSTN*) plays a negative role in regulating the number and size of skeletal myocytes. Additionally, it is reported that mutations of *MSTN* gene contribute to the double-muscling (DBM) phenomenon. Therefore, mutations of *MSTN* gene will always be one of the hot spots. A 5 bp indel (c. –120 ins) in the 5’ untranslated region (5’ UTR) of goat *MSTN* gene was reported to relate to the growth traits of goat. However, all the sample sizes were limited. Herein, this study firstly enlarged the sample size (n = 1074, Shaanbei White Cashmere goat) to uncover the indel location, as well as its association with growing performance. Based on the data, the 5 bp insertion mutation in goat *MSTN* gene was significantly associated with the body height, height at hip cross, and chest width index in SBWC (*p* < 0.05), hinted that this insertion could be assigned to an effective molecular marker for growth trait in goat rearing.

**Abstract:**

Myostatin (*MSTN*) gene, also known as growth differentiation factor 8 (*GDF8*), is a member of the transforming growth factor-beta super-family and plays a negative role in muscle development. It acts as key points during pre- and post-natal life of amniotes that ultimately determine the overall muscle mass of animals. There are several studies that concentrate on the effect of a 5 bp insertion/deletion (indel) within the 5’ untranslated region (5’ UTR) of goat *MSTN* gene in goats. However, almost all sample sizes were below 150 individuals. Only in Boer goats, the sample sizes reached 482. Hence, whether the 5 bp indel was still associated with the growth traits of goats in large sample sizes which were more reliable is not clear. To find an effective and dependable DNA marker for goat rearing, we first enlarged the sample sizes (n = 1074, Shaanbei White Cashmere goat) which would enhance the robustness of the analysis and did the association analyses between the 5 bp indel and growth traits. Results uncovered that the 5 bp indel was significantly related to body height, height at hip cross, and chest width index (*p* < 0.05). In addition, individuals with DD genotype had a superior growing performance than those with the ID genotype. These findings suggested that the 5 bp indel in *MSTN* gene are significantly associated with growth traits and the specific genotype might be promising for maker-assisted selection (MAS) of goats.

## 1. Introduction

Increasing need for mutton requires that certain growth-related traits of local goat breeds which are always of primary concern should be remarkably improved [1]. Early growth traits of animals, being regulated by multi-genes, as well as their interactions, play a vital role in influencing profitability in any meat-producing enterprise [2,3]. Thus, to establish optimal breeding programs, plenty of candidate genes, as well as their genetic variations should be appropriately used [4,5,6,7]. Maker-assisted selection (MAS) can satisfy the need to screen crucial genes and consider the relation between their polymorphisms and growth-related traits [8,9,10,11]. Herein, establishing a MAS system and performing candidate genes analyses will speed up the development of goat breeding [12,13,14,15].

Myostatin (*MSTN*) gene, also known as growth and differentiation factor 8 (*GDF8*), is highly conserved in many tissues (including the mammary gland) but most prominently in skeletal muscle [16]. *MSTN* exons encode for a 375 amino-acid (aa) latent protein that undergoes significant post-translational modification to become biologically active [17]. It is one of the predominant regulators that inhibits skeletal muscle growth especially during the process of pre-natal muscle development which completely determines the number of myofibers: Muscle precursor proliferation, myoblast proliferation, and differentiation [18,19]. The *MSTN* gene upregulates the cyclin-dependent kinase inhibitor p21 expression (Waf1, Cip1 signal pathway) and downregulates the Cdk2 protein which contributes to the arrest of myoblast in G (1)-phase of cell cycle. Ultimately, the deregulated of myoblast proliferation leads to muscular hyperplasia [20]. The deficiency of *MSTN* contributes to an increase in skeletal muscle mass known as double-muscling (DBM). The double-muscle phenomenon was firstly reported by Culley and Woodfall in 1807 [21]. The extensive breakthrough in understanding skeletal myogenesis was accomplished by delineating natural mutation within the *MSTN* causing double muscling phenotype in the Belgian Blue and Piedmontese Cattle breeds [22,23,24,25]. A similar phenotype can also be observed in the *MSTN* gene of knockdown animals. The first *MSTN*-deficient mice were reported to be 30% greater in muscle mass than heterozygous and wild-type normal mice [26,27]. With the lucubration of the above phenomenon, RNAi, molecular cloning, gene knockdown, as well as gene editing technology such as CRISPR/Cas9, TALEN, and ZFN were also applied to the goat sheep [28,29,30,31,32,33,34].

In addition, the *MSTN* mutations are necessary for hypertrophy, muscle mass, growth, and other traits, with lesser roles played by other gene variants [35]. Therefore, *MSTN* and its variation are becoming one of the hot spots for association analysis of growth traits and meat traits in the animal breeding. Numerous studies and researches reported that there were various SNPs and indels loci in both domestic and exotic commercial breeds of goat, sheep, cattle, pig, rabbit, as well as poultry [36]. Additionally, plenty of research focused on a 5 bp indel in 5’ UTR of *MSTN* gene within different goat breeds to uncover its distributions and effects on growth traits of goats [37]. A research detected distributions of the 5 bp indel in Black Bengal, Sirohi, Osmanabadi, Jakhrana, Jamunapari, Barbari, Marwari goat breeds [38]. Furthermore, several reports revealed that the 5 bp indel was associated with body length in Boer goats [39] and circumference of cannon bone, hip width, chest depth in backcrossed offsprings of Boer goats and Tangshan Dairy goats [40]. What is worth mentioning here are the relevant reports about the association between variations and growth traits that are limited in sample sizes [39,40].

As the main genetic source of cashmere in local goat industry of North China, the Shaanbei White Cashmere goat which is a hybrid goat between Liaoning Cashmere goat (♂) and Shaanbei Ziwuling Black goat (♀), has the resistance to the crude feed, cold, wind-blown sand and disease [41]. Additionally, it has superior qualities and quantities of cashmere than other cashmere goat breeds. However, growth traits of Shaanbei White Cashmere goats need improving. Rapid, accurate, as well as an effective method of MAS is presented in [42,43,44]. Herein, this study aimed at finding an effective and faithful DNA maker which could accelerate the process of selective breeding in goat rearing. A total of 1074 Shaanbei White Cashmere goats were analyzed, aiming at enhancing the robustness, as well as revealing the impact of the 5 bp indel on the growth performance. These results may provide references for further research on applying MAS to the goat industry.

## 2. Materials and Methods

All experiments implemented in this study were approved by the International Animal Care and Use Committee of the Northwest A&F University (IACUC-NWAFU) and fully followed local animal welfare guidelines, laws, and policies. Additionally, all experimental processing in this study were approved by the International Animal Care and Use Committee (IACUC) of the Northwest A&F University (protocol number NWAFAC1008) on animal experiment, including sample collection, were performed with the guidelines of the ethics commission.

### 2.1. Sample Collection, Phenotypic Traits Recording, and Genomic DNA Isolation

A total of 1074 ear tissue samples (adults n = 472; kids, n = 602) were collected randomly from the Shaanbei White Cashmere goat breeding farm. Nine growth trait records (some body weight records were missing) were provided by the agricultural technical station of Yulin city, Shaanxi Province. DNA was extracted using a high-salt extraction method [45,46]. After being assayed by a Nano-drop 1000 spectrophotometer (Thermo Scientific, Waltham, MA, USA), all DNA samples were diluted to the same standard of 20 ng/μL and were stored at 4 °C temporarily. Additionally, 30 DNA samples of Shaanbei White Cashmere goat were mixed in a genomic DNA pool for a polymerase chain reaction (PCR) analysis [47].

### 2.2. Primer Design, PCR Amplification, and Genotyping

Using the Primer Premier 6.0 (Premier, Canada), the primer pair (F: 5’-ACTGGTGTGGCAAGTTGTCTCT-3’; R: 5’-TTCCTTCTGCTCGCTGTTCTCA-3’) for amplification of indel loci in 5’ UTR was designed based on the goat *MSTN* gene sequence (NC_030809.1) and the Ensemble Indel-database (http://apr2019.archive.ensembl.org/index.html/indel). Assays were performed by touch-down PCR in a 13 μL volume, containing 6.5 μL 2 × mix, 0.2 μL each of forward and reverse primers, 0.5 μL genomic DNA (20 ng/μL), and 5.6 μL ddH_2_O. The PCR protocol was as below: Initial denaturation for 5 min at 95 °C; followed by 18 cycles of denaturation for 30 s at 95 °C, annealing for 30 s at 68 °C (with a decrease of 1 °C per cycle), extension for 15 s at 72 °C; another 38 cycles of 30 s at 95 °C, 30 s at 50 °C, and 15 s at 72 °C, and a final extension for 5 min at 72 °C, with subsequent cooling to 12 °C. Using 3.5% agarose gel electrophoresis, spoting 5 μL and running for 1.5 h, the 5 bp indel of goat *MSTN* gene was examined via electrophoresis at 120 V voltage. Then, genotypes were identified when combined with sequencing.

### 2.3. Statistical Analyses

To explore the genetic diversity of the indel variants in the investigated goat population, we calculated genetic parameters. The genotype and allele frequencies reflect the genetic composition of the indel variant in the tested goat population. We calculate the population genetic diversity indices, including homozygosity (Homo), heterozygosity (Hetero), and polymorphism information content (PIC) in Nei’s methods [48]. Homo and Hetero (Homo + Hetero = 1) are measures of genetic variation of a population. PIC is an indicator of polymorphism. Based on PIC values, the genetic variations classified as low genetic diversity (PIC < 0.25), medium genetic diversity (0.25 < PIC < 0.5), and high genetic diversity (PIC > 0.5) [49]. Cause there were only two genotypes (DD and ID), the independent-samples t-test was used in the study [50].

## 3. Results

### 3.1. Identification and Genetic Parameter Analysis of Indel Variation

A 5 bp indel (c. −120 ins) in the 5’ UTR of goat *MSTN* gene was detected. There were two different genotypes: Homozygote deletion type (deletion/deletion: DD, 206 bp); heterozygote type (insertion/deletion: ID, 211 bp and 206 bp) (Figure 1).

The genotype and allele frequencies, as well as other genetic parameters, associated with the *MSTN* indel loci were calculated to determine the genotype distribution among Shaanbei White Cashmere goats (Table 1). The data indicated that “D” allele (0.943) of the 5 bp indel was more frequent than “I” allele (0.057). Based on PIC values, the 5 bp locus had a low genetic diversity (PIC = 0.055).

Published reports also performed the genetic parameter analyses in several different goat breeds (Appendix A). The “D” alleles were totally more frequent than “I” alleles while different goat breeds had different genetic diversities.

### 3.2. Associations between Indel Variations and Growth Traits

From (Table 2 and Appendix A, Figure 2), association analyses showed that the 5 bp indel was significantly associated with the body height, height at hip cross and chest width index (*p* < 0.05) from 602 kids. Interestingly, the “DD” genotype was superior than “ID” genotype in the body height and the height at hip cross which was reverse to the chest width index.

In addition, we did a correlation analysis between growth traits of which “DD” genotype had a superior performance than “ID” genotype (BH, HHC, CC, BW). Results manifested that analyzed growth traits were positively correlated with each other (Appendix A).

There is no significance between the 5 bp indel and growth traits in adults. However, the individual with “DD” had a superior performance than those with “ID” in body height, height at hip cross, hip width, chest width, cannon circumference, chest depth, body weight, heart girth traits which was coincident with the body height and height at hip cross of the kids.

## 4. Discussion

In this study, the relation between the 5 bp indel and growth traits of SBWC goats was detected. Genetic parameter analysis confirmed that the 5 bp indel was at Hardy–Weinberg equilibrium. Additionally, the allelic frequency of “D” is higher than “I” which was consistent with several exotic goat breeds such as Black Bengal, Sirohi, Osmanabdi, Maerhoz, and domestic goat breeds such as Boer, Matou goats. Interestingly, the “DD” genotype was the dominant genotype and had a superior phenotype than the “ID” genotype in the above goat breeds [38]. While the circumstance in Nubi goat breeds was different. It is reported that three genotypes of the 5 bp indel were found in the Nubi goat and the “I” allele, as well as the “ID” genotype was dominant in Nubi goat breed [39]. Additionally, the 5 bp indel within the Shaanbei White Cashmere goat was at low genetic diversity, while there were low, medium, and high genetic diversity in other domestic and exotic commercial goat breeds, respectively. In detail, in Black Bengal, Osmanabadi, Jakhrana, Jamunapari, Barbari, Marwari, Markhoz, Matongoats goat breeds, the 5 bp indel had low genetic diversities (0 < PIC < 0.164). In Boer, Haimen, Nubi, South group, North group, Foreign group, as well as the backcrossed offspring, the PIC values ranged from 0.279 to 0.375 which illustrated that the 5 bp indel among these breeds had medium genetic diversities. Only in Sirohi goats, the 5 bp indel had high genetic diversity (PIC = 1.133) [38,39]. Next, the association between different genotypes and growth traits was analyzed. The circumstance was that the 5 bp indel was distinctly related with body height, height at hip cross, and chest depth index in kids. It could be explained that the *MSTN* gene mutations gave rise to hyperplasia and hypertrophy which leaded to an increasing in muscle mass [51]. Additionally, we did a correlation analysis between growth traits which had a consistent phenotype with the body height and height at hip cross. It revealed that all growth traits had a positive correlation (*p* < 0.05) with each other. Hence, we speculate that the 5 bp indel can affect other growth traits of Shaanbei White Cashmere goats.

The *MSTN* gene plays a crucial role in the development of domestic animals due to its key function in muscularity [52]. Its exons code for a 375 amino-acid (aa) latent protein which will become biologically active undergoing significant post-translational modification. Firstly, through formatting the disulphide bonds, the polypeptide undergoes intracellular homodimerization which cleaved the polypeptide to form the N-terminal propeptide region and the C-terminal mature region. The 12 kDa C-terminal mature fragment of *MSTN* gene initiates an intracellular signaling cascade according to bind and activates the activin type II receptor at the cell surface (ActRIIB and to a lesser extent ActRIIA). Subsequent autophosphorylation of ActRIIB gives rise to recruiting and activating the low affinity type I receptor for activin ALK-4 or ALK-5. Moreover, activated type I receptor kinase phosphorylates the transcription factors Smad2 and Smad3 which allows them to interact with Smad4 (co-Smad) and to translocate to the nucleus to activate target gene transcription. Importantly, the activation of the *MSTN* receptor also inhibits Akt (protein kinase B) activity which displays a decisive role in muscle protein synthesis and cell proliferation [53]. On the contrary, the deletion of goat *MSTN* gene will enhance the muscle protein synthesis and cell proliferation which contribute to an aggrandizement on the mass of muscle, then gain the body weight. Since the body height had a positive correlation with the body weight [54], the deletion of the *MSTN* gene was considered to have a positive effect on body weight. Additionally, we also hypothesized that the mutations of goat *MSTN* gene were related to the phenotype of other growth traits.

Briefly, the 5 bp indel of 5’ UTR within the goat *MSTN* was found to show different genotypic and allelic distributions in Shaanbei White Cashmere goats which manifest that it can be potential DNA makers for a further step of selecting high-quality individuals with MAS in breeding.

## 5. Conclusions

The 5 bp insertion mutation in the 5‘ UTR within the *MSTN* gene was detected in SBWC goats. Additionally, the “D” allele was the dominant allele in this population which displayed a low genetic diversity. Interestingly, the “DD” genotype was the dominant genotype. Furthermore, association analysis showed that the 5 bp indel was significantly associated with the body height, height at hip cross, and chest width index in SBWC (*p* < 0.05), hinting that this insertion could be assigned to an effective molecular marker for growth trait in goat rearing.

## Figures and Tables

**Figure 1 animals-10-00168-f001:**
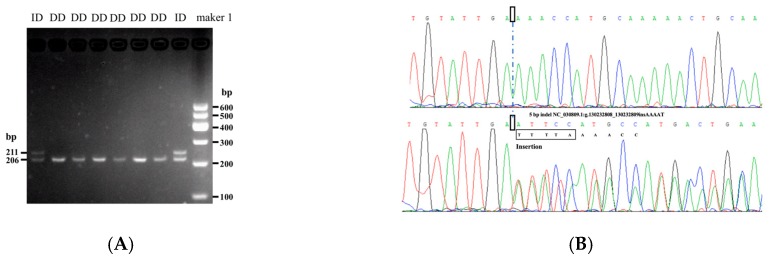
Agarose electrophoresis of *MSTN* 5 bp indel (**A**), and sequencing chromas for the 5 bp indel in the *MST**N* gene (**B**). Sequencing chromas showed homozygotic deletion type (DD) and heterozygote type (ID).

**Figure 2 animals-10-00168-f002:**
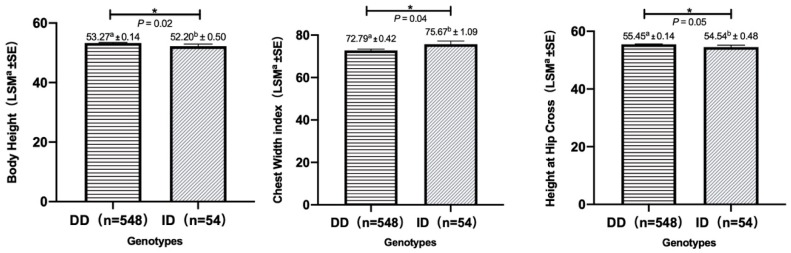
Association of the 5 bp indel within the *MSTN* gene with body height, height at hip cross, and chest width index in kids of SBWC goats.

**Table 1 animals-10-00168-t001:** Genetic parameters for the 5 bp indel of the SBWC goat *MSTN (Myostatin)* gene.

Sample Sizes	Frequencies		Homo	Hetero	Ne	PIC	HWE *p*-Value
	Genotypes	Alleles					
N = 1074	DD 0.890	D 0.943	0.896	0.104	1.116	0.098	0.055
	ID 0.110	I 0.057					
	II 0.000						

Note: HWE: Hardy–Weinberg equilibrium; Homo: Homozygosity; Hetero: Heterozygosity; Ne: Effective allele numbers; PIC: Polymorphism information content; SBWC: Shaanbei White Cashmere goat.

**Table 2 animals-10-00168-t002:** Relationship between the 5 bp indel locus within the *MSTN* gene and the growth traits in kids of SBWC goats (LSM ^a^ ± SE) (*p* < 0.05).

Growth Traits	Observed Genotypes (LSM ^a^ ± SE)		*p*-Values
	DD	ID	
Body Height (cm)	53.27 ^a^ ± 0.14 (n = 548)	52.20 ^b^ ± 0.50 (n = 54)	0.02
Height at Hip Cross (cm)	55.45 ^a^ ± 0.14 (n = 548)	54.54 ^b^ ± 0.48 (n = 54)	0.05
Body Length (cm)	61.70 ± 0.20 (n = 548)	60.73 ± 0.48 (n = 54)	0.13
Hip Width (cm)	12.85 ± 0.07 (n = 548)	12.96 ± 0.30 (n = 54)	0.72
Chest Width (cm)	18.48 ± 0.11 (n = 548)	19.10 ± 0.36 (n = 54)	0.09
Cannon bone Circumference (cm)	7.38 ± 0.03 (n = 548)	7.32 ± 0.10 (n = 54)	0.47
Chest Depth (cm)	25.44 ± 0.11 (n = 548)	25.25 ± 0.31 (n = 54)	0.54
Body Weight (kg)	36.22 ± 0.93 (n = 73)	35.70 ± 2.36 (n = 11)	0.84
Heart Girth (cm)	72.11 ± 0.31 (n = 548)	72.30 ± 0.94 (n = 54)	0.92
Body Trunk index	118.36 ± 1.77 (n = 548)	119.07 ± 1.29 (n = 54)	0.90
Body Length index	115.98 ± 0.47 (n = 548)	116.89 ± 1.42 (n = 54)	0.57
Heart Girth index	135.67 ± 0.69 (n = 548)	139.27 ± 2.41 (n = 54)	0.12
Cannon bone Circumference index	13.89 ± 0.07 (n = 548)	14.08 ± 0.22 (n = 54)	0.43
Chest Width index	72.79 ^a^ ± 0.42 (n = 548)	75.67 ^b^ ± 1.09 (n = 54)	0.04
Hip Width index	147.57 ± 2.97 (n = 548)	149.50 ± 2.89 (n = 54)	0.84

Note: Cells with different letters. ^a,b^ differed significantly (*p* < 0.05).

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
