# Peer review of "Myostatin (MSTN) Gene Indel Variation and Its Associations with Body Traits in Shaanbei White Cashmere Goat"

_animals, 2020, doi:10.3390/ani10010168_

Round 1

Reviewer 1 Report

This paper is of great interest to Animals. It is very well designed and performed all its development is correct Only formal corrections must be recommended to improve the quality of the manuscript.

The following suggestions try to correct the few misunderstanding found in the text:

General comments:

Title must be changed to “3 Myostatin (MSTN) Gene Indel Variation and its Associations with 4 Body Traits in Shaanbei White Cashmere goat ” in order to simplify it and to clarify its relation to the contents. Really the works are developed in a sigle local breed and not in the whole specie.

Key words must not be included in the title with a view to increase the visibility of the paper.

Conclusions are almost an extension of the results they must be enriched with other sentences regarding to the relevance of these findings in the sector. Also the results must be concretized on the  Shaanbei White Cashmere goat and not extended to the whole specie.

Text must be carefully reviewed, even the English has a general good quality, there are several errors in the manuscript, for instance:

Line 131.- … correltion between grow… replace to correlation

Other minos suggestions:

Line 77 .- Singh et al must include the year of publication, and al must shows a point al.

Line 83.- … worth mentioning here are the relevant reports about… which reports?, references are needed

Lines 147- 155 .- All of this big paragraph is part of the discussion, please relocate

In conclusion I may recommend the acceptance of this paper after a minor review

Author Response

A cover letter with responses to the reviewers' comments and editorial comments on Animals-665915

Dear Editor and anonymous reviewers,

We greatly appreciate the anonymous reviewers for your careful review and constructive comments of our manuscript (ID: animals-665915). We have studied comments carefully and tried our best to revise this manuscript, and we hope that the revision can meet with your approval.

Here, we have listed the point-by-point responses to your detailed comments and suggestions (with red). As follows:

[Question 1] 

Title must be changed to “3 Myostatin (MSTN) Gene Indel Variation and its Associations with 4 Body Traits in Shaanbei White Cashmere goat” in order to simplify it and to clarify its relation to the contents. Really the works are developed in a sigle local breed and not in the whole specie.

[Response 1]

Thank you very much for your valuable advice. The title has been changed to “Myostatin (MSTN) Gene Indel Variation and its Associations with Body Traits in Shaanbei White Cashmere goat”.

[Question 2]

Key words must not be included in the title with a view to increase the visibility of the paper.

[Response 2]

Thanks for your careful suggestion. We have changed the key words to “goat, MTSN gene, 5-bp indel, growth, correlation”.

[Question 3]

Conclusions are almost an extension of the results they must be enriched with other sentences regarding to the relevance of these findings in the sector. Also, the results must be concretized on the Shaanbei White Cashmere goat and not extended to the whole specie.

[Response 3]

Thank you. The conclusion has been enriched with some details about the study.

Specific modifications are as follows:

“The 5-bp insertion mutation in the 5‘UTR within MSTN gene was detected in SBWC goats. Additionally, the “D” allele was the dominant allele in this population which displayed a low genetic diversity. Interestingly, the “DD” genotype was the dominant genotype. Furthermore, association analysis showed that the 5- bp indel was significantly associated with the body height, height at hip cross and chest width index in SBWC (P < 0.05), hinted that this insertion could be assigned to an effective molecular marker for growth trait in goat rearing.”

[Question 4]

Text must be carefully reviewed, even the English has a general good quality, there are several errors in the manuscript, for instance:

Line 131 … correltion between grow… replace to correlation

[Response 4]

Thank you for your careful proposal. The spelling mistake has been revised to “correlation”.

[Question 5]

Line 77: Singh et al must include the year of publication, and al must shows a point al.

[Response 5]

Thanks. The format of the Table S1 has been carefully adjusted. Additionally, the year of publication was also added in the table.

[Question 6]

Line 83: … worth mentioning here are the relevant reports about… which reports? references are needed

[Response 6]

Feeling indebted for your opinion. The references have been suppled. Specific revisions are as follows:

“What is worth mentioning here are the relevant reports about the association between variations and growth traits are limited in sample sizes [39-40]”.

[39] Zhang, C.Y.; Liu, Y.; Xu, D.Q.; Wen, Q.Y.; Li, X.; Zhang, W.M.; Yang, L.G. Polymorphisms of myostatin gene ;(MSTN) in four goat breeds and their effects on Boer goat growth performance. Molecular Biology Reports. 2012, 39, 3081-3087.

[40] Li, X.L.; Liu, Z.Z.; Zhou, R.Y.; Zheng, G.R.; Gong, Y.F.; Li, L.H. Deletion of TTTTA in 5’UTR of goat MSTN gene and its distribution in different population groups and genetic effect on body weight at different ages. Frontiers of Agriculture in China. 2008, 2, 103-109.

[Question 7]

Lines 147-155: All of this big paragraph is part of the discussion, please relocate

[Response 7]

Deeply Grateful for your advice. The part has been merged into discussion. It was modified to:

“Additionally, the 5-bp indel within Shaanbei White Cashmere goat was at low genetic diversity, while there were low, medium and high genetic diversity in other domestic and exotic commercial goat breeds respectively. In details, in Black Bengal, Osmanabadi, Jakhrana, Jamunapari, Barbari, Marwari, Markhoz, Matongoats goat breeds, the 5-bp indel had low genetic diversities (0 < PIC < 0.164). In Boer, Haimen, Nubi, South group, North group, Foreign group as well as the backcrossed offspring, the PIC values ranged from 0.279 to 0.375 which illustrated that the 5-bp indel among these breeds had medium genetic diversities. Only in Sirohi goats, the 5-bp indel had high genetic diversity (PIC=1.133)”.

With best regards!

Sincerely Yours,

Yi Bi (biyi0312@163.com),

Xianyong Lan (lanxianyong79@126.com),

Xiaoyue Song (songxiaoyue@yulinu.edu.cn) (corresponding author),

Chuanying Pan (panyu1980@126.com), et al.

College of Animal Science and Technology,

Northwest A&F University, Yangling, Shaanxi 712100, China

Reviewer 2 Report

The manuscript by Bi et al, describes the identification of indel variation in caprine MSTN gene, and investigates association between indel variation in caprine MSTN gene and variation in growth traits. The writing quality of the paper is adequate, and aims are good. However, my biggest concern is statistical analyses. In the model, gender, sire and birth rank are not considered. Therefore, the association results are incredibility although having larger sample sizes. 

When descripting variation or SNPs in a gene,  a coding DNA reference sequence is preferred. For example "c." . Please refer to HGVS nomenclature (http://varnomen.hgvs.org/).

Lines 148-155: belong to discussion.  

Lines 169-183: Since age has been fitted as a fixed factor in the model, why not included  kids and ewes together in the association analyses. Merging kids and ewes into a model is suggested. 

Discussions: compare correlation results in the study to those in other goat breeds.

Author Response

A cover letter with responses to the reviewers' comments and editorial comments on Animals-665915

Dear Editor and anonymous reviewers,

We greatly appreciate the anonymous reviewers for your careful review and constructive comments of our manuscript (ID: animals-665915). We have studied comments carefully and tried our best to revise this manuscript, and we hope that the revision can meet with your approval.

Here, we have listed the point-by-point responses to your detailed comments and suggestions (with red). As follows:

[Question 1]

The writing quality of the paper is adequate and aims are good. However, my biggest concern is statistical analyses. In the model, gender, sire and birth rank are not considered. Therefore, the association results are incredibility although having larger sample sizes. 

[Response 1]

Greatly appreciated for your suggestion.

In this study, the adjusted linear model with fixed effects was used to deal with the relationships between genotypes and growth traits of 1074 SBWC goats. The adjusted model included fixed effects of marker genotype, birth year, sire, sex, breed and random effects of animal.

Linear Model: Yijkl = μ + Ai + Sj + Gk + BYl + Eijkl, where Yijkl was the trait measured on each of the ijklth animal; μ the overall mean; Ai is the evaluated on the ith level of the fixed factor age; Sj the type of the jth gender; Gk the type of the kth gentotype; BYl, the type of birth year, and Eijkl was the random error.

An effect associated with sex and season of birth (spring versus fall), sire was not into linear model, as the previous study indicated that these effects did not have significant influences on variability of traits in female populations (Lan et al., 2007). Additionally, all analyzed goats were female SBWC goats and there was no kinship between them. Therefore, the reduced model used in the final analysis is as follows: Yijkl = μ + Ai + Gk + Eijkl.

Reference:

Lan, X.Y.; Pan, C.Y.; Chen, H.; Zhang, C.L.; Li, J.Y.; Zhao, M.; Lei, C.Z.; Zhang, A.L.; Zhang, L. An AluI PCR-RFLP detecting a silent allele at the goat POU1F1 locus and its association with production traits. Small Ruminant Research. 2007, 73, 8-12.

[Question 2]

When descripting variation or SNPs in a gene, a coding DNA reference sequence is preferred. For example, "c." Please refer to HGVS nomenclature (http://varnomen.hgvs.org/).

[Response 2]

Thanks for your advice. Given that the 5-bp indel was not in the coding region of goat MSTN gene, so we described it as NC_030809.1: g.130232808_130232809insAAAAT.

[Question 3]

Lines 148-155: belong to discussion.  

[Response 3]

Thank you very much for your suggestion. This part has been put into discussion.

Specific modifications are as follows:

“In details, in Black Bengal, Osmanabadi, Jakhrana, Jamunapari, Barbari, Marwari, Markhoz, Matongoats goat breeds, the 5-bp indel had low genetic diversities (0 < PIC < 0.164). In Boer, Haimen, Nubi, South group, North group, Foreign group as well as the backcrossed offspring, the PIC values ranged from 0.279 to 0.375 which illustrated that the 5-bp indel among these breeds had medium genetic diversities. Only in Sirohi goats, the 5-bp indel had high genetic diversity (PIC=1.133)”.

[Question 4]

Lines 169-183: Since age has been fitted as a fixed factor in the model, why not included kids and ewes together in the association analyses. Merging kids and ewes into a model is suggested. 

[Response 4]

Greatly appreciated with your pertinent and careful suggestion.

Considering your suggestion carefully, we would like to discuss our opinion with yo:

In the last manuscript, we fitted the age of SBWC goat into the model. However, given that goats growth varied greatly at different period, so we removed the age factor from the model and divided the analyzed population into two groups: kids (n=602) and adults (n=472) to do association analyze respectively.

In this study, the adjusted linear model with fixed effects was used to deal with the relationships between genotypes and growth traits of 1074 SBWC goats. The adjusted model included fixed effects of marker genotype, birth year, sire, sex, breed and random effects of animal.

Linear Model: Yjkl = μ+ Sj + Gk + BYl + Ejkl, where Yjkl was the trait measured on each of the jklth animal; μ the overall mean; Sj the type of the jth gender; Gk the type of the kth gentotype; BYl, the type of birth year, and Ejkl was the random error.

An effect associated with sex and season of birth (spring versus fall), sire was not into linear model, as the previous study indicated that these effects did not have significant influences on variability of traits in female populations (Lan et al., 2007). Additionally, all analyzed goats were female SBWC goats and there was no kinship between them. Therefore, the reduced model used in the final analysis is as follows: Yjkl = μ+ Gk + Ejkl.

Reference:

Lan, X.Y.; Pan, C.Y.; Chen, H.; Zhang, C.L.; Li, J.Y.; Zhao, M.; Lei, C.Z.; Zhang, A.L.; Zhang, L. An AluI PCR-RFLP detecting a silent allele at the goat POU1F1 locus and its association with production traits. Small Ruminant Research. 2007, 73, 8-12.

 [Question 5]

Discussions: compare correlation results in the study to those in other goat breeds.

[Response 5]

Pertinent opinion as it is. We have already added some comparison into the discussion as follows:

“In this study, the relation between the 5-bp indel and growth traits of SBWC goats was detected. Genetic parameter analysis confirmed that the 5-bp indel was at Hardy-Weinberg equilibrium. Additionally, the allelic frequency of “D” is higher than “I” which was consistent with several exotic goat breeds like Black Bengal, Sirohi, Osmanabdi, Maerhoz and domestic goat breeds like Boer, Matou goats. Interestingly, the “DD” genotype was the dominant genotype and had a superior phenotype than “ID” genotype in above goat breeds [38]. While the circumstance in Nubi goat breeds was different. It is reported that three genotypes of the 5-bp indel were found in the Nubi goat and the “I” allele as well as the “ID” genotype was dominant in Nubi goat breed [39].”

With best regards!

Sincerely Yours,

Yi Bi (biyi0312@163.com),

Xianyong Lan (lanxianyong79@126.com),

Xiaoyue Song (songxiaoyue@yulinu.edu.cn) (corresponding author),

Chuanying Pan (panyu1980@126.com), et al.

College of Animal Science and Technology,

Northwest A&F University, Yangling, Shaanxi 712100, China

Reviewer 3 Report

The manuscript entitled “Investigation of indel variation within the goat myostatin (MSTN) gene and its associations with body measurement traits” makes an analysis of an indel for the MSTN gene, remarking the greater sample size of the analysis, which contributes to more statistically robust results. The findings from the study are most likely of interest of readers of Animals. However, there are some clarifications needed in order for the manuscript to be suitable for publication.

Line 19 to the double-muscling

Line 20 …gene will always be…

Line 34 First

Line 35 I suggest modifying “enhance credibilidty” for something more appropriate. Like, enhancing the robustness of the analysis –statistically speaking- .

Line 37.  Skip “for the first time”

Line 39 suggest and are instead of suggested and was

Line 46 should be remarkably improved.

Line 54 what do you mean with MSTN? The gene or the protein? please clarify this phrase

Line 60. MSTN upregulates p21.  Please clarify what is p21 (a cyclin-dependant kinase inhibitor) and if you could say to which metabolic pathway it is involved with it would explain better the role of MSTN for non-expert readers.

Line 63 First ……. Cooley & Woodfall

Line 65 Belgian Blue and Piedmontese cattle breeds

Lines 69 and 70. Authors refer to gene-editing technologies in different species, but somehow in the given references (24-29) they just mention goat and sheep manuscripts

Line 74 please change “a good deal” you may use “numerous studies” instead

Line 75 what do authors mean with “import” are they referring to commercial and local breeds?.... also in lines 212 and 213

Line 77 Singh et al.   I don´t fully get this statement. What is Black Bengal… etc, are those breeds? And, of which species and distributions of what? The indel? Please rephrase

Line 82 statistical significance

Lines 85 to 87, replace “goat” for breeds, or breeds of goats

Line 87   please modify “diseases and so on…” this is no proper scientific writing, and give references

Line 89 breed

Line 92-93  Aren´t the authors too pretentious with the statement of “potential theories”?

Line 104 skip “for DNA experiment”

Statistical analysis

Lines 123-124 disagree with this statement. Genetic diversity analyses do not define the genetic structure f a mutation, those are different concepts: genetic diversity and genome structure.

Lines 130 to 135 It is not clear that if the authors did an association analysis and then they used the SPSS to prove its statistical significance. In general, this part needs to improve its redaction, as it is highly relevant for the manuscript and it is confusing as it is.

The genetic diversity indices, as referred by the authors include Ho and He for homozygosity and heterozygosity. These acronyms are confusing. population genetic analyses use these acronyms to define Observed heterozygosis (Ho) and Expected Heterozygosis (He). Please change the acronyms in the manuscript.

Lines 147-148 published reports, is part of the discussion.

Line 149 please rephrase “in details”… and again skip “goats” in each breed, instead, you can mention “goat breeds” at the end.

Table 2 Can the authors explain why is it that the sample number for all phenotypes is 548 in the homozygotes, but just 73 on body weight? Bodyweight is a relevant trait for the MSTN gene, and… if the phenotypes registered for this trait where just 73 it should be stated somewhere in material and methods. Same observation in Appendix 2

Appendix 1  is impossible to read as it is, the numbers are not in the same line. Please accommodate the columns properly so the table can be read.

198 “being indel markers extensively used …..”

In general, in the whole manuscript: avoid tendentious words that may establish decisive judgments, such as: absolutely (line 50), definitely (l.52), dramatically (l.200), remarkably (l.239) etc.

Line 200 is a major gene which/that regulate

Line 201 rephrase “a mass of research”

In the results part, the authors do not give the results of the statistical model applied, they refer to appendix 3 for the correlation matrix but previously I miss results of one of the main analyses of the manuscript.

Appendix 3. Youth  and Cannon bone circumference

From lines 198 to 208 the authors give a mixture that goes from an introduction and conclusions in between. I would consider removing this, as it is repetitive information or discussing previous studies regarding the results obtained in this study.

Author Response

A cover letter with responses to the reviewers' comments and editorial comments on Animals-665915

Dear Editor and anonymous reviewers,

We greatly appreciate the anonymous reviewers for your careful review and constructive comments of our manuscript (ID: animals-665915). We have studied comments carefully and tried our best to revise this manuscript, and we hope that the revision can meet with your approval.

Here, we have listed the point-by-point responses to your detailed comments and suggestions (with red). As follows:

[Question 1]

Line 19: to the double-muscling

[Response 1]

Greatly appreciated with your recommendation. The “double muscling” was already replaced by “double-muscling”.

[Question 2]

Line 20: …gene will always be…

[Response 2]

Thank you. The sentence was already modified to “Therefore, mutations of MSTN gene will always be one of the hot spots."

[Question 3]

Line 34 First

[Response 3]

Thank you for your careful proposal. The “Firstly” was already replaced by “First”.

[Question 4]

Line 35 I suggest modifying “enhance credibilidty” for something more appropriate. Like, enhancing the robustness of the analysis –statistically speaking.

[Response 4]

Deeply indebted to your suggestion. Specific modification are as follows:

“To identify the 5-bp indel as well as its effects on growth traits, we first enlarged the sample sizes (n=1074, Shaanbei White Cashmere goat) to enhance the robustness of the analysis.”

[Question 5]

Line 37.  Skip “for the first time”

[Response 5]

Greatly thankful. We have already deleted “for the first”.

[Question 6]

Line 39 suggest and are instead of suggested and was

[Response 6]

Thank you very much for your meticulous suggestion. We have already revised the past tense to the present tense.

[Question 7]

Line 46 should be remarkably improved.

[Response 7]

Thanks. The word order was already changed to “which are always of primary concern should be remarkably improved”.

[Question 8]

Line 54 what do you mean with MSTN? The gene or the protein? please clarify this phrase

[Response 8]

In line 54, the “MSTN” meant the “MSTN” gene. We have clarified the phrase.

[Question 9]

Line 60. MSTN upregulates p21.  Please clarify what is p21 (a cyclin-dependant kinase inhibitor) and if you could say to which metabolic pathway it is involved with it would explain better the role of MSTN for non-expert readers.

[Response 9]

Deeply appreciated with your pertinent advice. We have enriched this part with the metabolic pathway which the p21 being involved with. Specific modifications are as follows:

MSTN gene upregulates the cyclin- dependent kinase inhibitor p21 expression (Waf1, Cip1 signal pathway) and downregulates the Cdk2 protein which contributes to the arrest of myoblast in G (1) - phase of cell cycle. Ultimately, the deregulated of myoblast proliferation leads to muscular hyperplasia”

[Question 10]

Line 63 First ……. Cooley & Woodfall

[Response 10]

Thank you. The “Cooley” was modified to “Cooley & Woodfall”.

[Question 11]

Line 65 Belgian Blue and Piedmontese cattle breeds

[Response 11]

Grateful for your carefully suggesting. We have already revised as you suggested.

[Question 12]

Lines 69 and 70: Authors refer to gene-editing technologies in different species, but somehow in the given references (24-29) they just mention goat and sheep manuscripts

[Response 12]

Deeply appreciated with your careful advice. We modified the sentence to “With the lucubration of the above phenomenon, RNAi, molecular cloning, gene knockdown as well as gene editing technology such as CRISPR/Cas9, TALEN, ZFN were also applied to the goat, sheep”.

[Question 13]

Line 74: please change “a good deal” you may use “numerous studies” instead

[Response 13]

Thanks. “a good deal” was already changed to “numerous studies”.

[Question 14]

Line 75: what do authors mean with “import” are they referring to commercial and local breeds? also in lines 212 and 213

[Response 14]

In the article, the “import” was replaced by “domestic” which meant the commercial and local goat breed.

[Question 15]

Line 77 Singh et al.   I don´t fully get this statement. What is Black Bengal… etc, are those breeds? And, of which species and distributions of what? The indel? Please rephrase

[Response 15]

The Black Bengal, Sirohi, Osmanabadi, Jakhrana, Jamunapari, Barbari, Marwari are all goat breeds. Moreover, “distributions” meant the distribution of the 5-bp indel in these goat breeds above.

Line 77 was carefully modified as follows: “A research detected distributions of the 5-bp indel in Black Bengal, Sirohi, Osmanabadi, Jakhrana, Jamunapari, Barbari, Marwari goat breeds”.

[Question 16]

Line 82: statistical significance

[Response 16]

Thanks for your proposal. We have already revised as you recommended.

[Question 17] Lines 85 to 87: replace “goat” for breeds, or breeds of goats

[Response 17]

Grateful for your advice. We replaced “goat” for “breed”.

[Question 18]

Line 87: please modify “diseases and so on…” this is no proper scientific writing, and give references

[Response 18]

Thank you for your suggestion. The sentence was modified and the reference was added.  Specific modification was as follows: “the Shaanbei White Cashmere goat which is a hybrid goat between Liaoning Cashmere goat (♂) and Shaanbei Ziwuling Black goat (♀), has the resistance to the crude feed, cold, wind-blown sand and disease [41]”

[41] Liu, A.Y. Progress in the production performance of Shaanbei white cashmere goat. Chine Herbivore Science. 2018, 27, 449-455.

[Question 19]

Line 89: breed

[Response 19]

Thanks. The sentence was modified to “The extensive breakthrough in understanding skeletal myogenesis was accomplished by delineating natural mutation within the MSTN causing double muscling phenotype in the Belgian Blue and Piedmontese Cattle breeds”.

[Question 20]

Line 92-93 Aren´t the authors too pretentious with the statement of “potential theories”?

[Response 20]

Greatly appreciated with your advice. We have already changed “potential theories” to “references”.

[Question 21]

Line 104: skip “for DNA experiment”

[Response 21]

Greatly thanks for your suggestion. The “for DNA experiment” has been deleted.

[Question 22]

Lines 123-124 disagree with this statement. Genetic diversity analyses do not define the genetic structure f a mutation, those are different concepts: genetic diversity and genome structure.

[Response 22]

Deeply indebted to your suggestion. We confused the definition of “genetic structure” before. After accessing to the information, we modified the sentence as follows:

“To explore the genetic diversity of the indel variants in the investigated goat population, we calculated genetic parameters.”

[Question 23]

Lines 130 to 135: It is not clear that if the authors did an association analysis and then they used the SPSS to prove its statistical significance. In general, this part needs to improve its redaction, as it is highly relevant for the manuscript and it is confusing as it is.

[Response 23]

Greatly appreciated with your pertinent and careful suggestion.

Considering your suggestion carefully, we would like to discuss our opinion with you:

In this section, we first described the linear model and then explained why the independent sample test was used in SPSS. It might contribute to a confusing understanding about the association analyze. In fact, we used the SPSS to do the association analysis. To make it more precise about how the analytical model is applied, we adjusted this part to:

“To explore the genetic diversity of the indel variants in the investigated goat population, we calculated genetic parameters. The genotype and allele frequencies reflect the genetic composition of the indel variant in the tested goat population. We calculate the population genetic diversity indices, including homozygosity (Homo), heterozygosity (Hetero) and polymorphism information content (PIC) in Nei’s methods [48]. Homo and Hetero (Homo + Hetero = 1) are measures of genetic variation of a population. PIC is an indicator of polymorphism. Based on PIC values, the genetic variations classified as low genetic diversity (PIC < 0.25), medium genetic diversity (0.25 < PIC < 0.5) and high genetic diversity (PIC > 0.5) [49]. For establishing the influence of different parameters on growth traits as well as the correltion between growth traits, associations between the indel and growth traits were analyzed using a general linear model: Yjkl = μ+ Gk + Ejkl, where Yjkl was the trait measured on each of the jklth animal; μ the overall mean; Gk the type of the kth gentotype, and Ejkl was the random error”.   [Question 24]

The genetic diversity indices, as referred by the authors include Ho and He for homozygosity and heterozygosity. These acronyms are confusing. population genetic analyses use these acronyms to define Observed heterozygosis (Ho) and Expected Heterozygosis (He). Please change the acronyms in the manuscript.

[Response 24]

Thanks a lot for your suggestion. The “Homo” and “Hetero” were used to replace “Ho” and “He” respectively. 

[Question 25]

Lines 147-148 “published reports” is part of the discussion.

[Response 25]

Thanks a lot. We have already put this part together with discussion. It was changed as follows:

“Additionally, the 5-bp indel within Shaanbei White Cashmere goat was at low genetic diversity, while there were low, medium and high genetic diversity in other domestic and exotic commercial goat breeds respectively. In details, in Black Bengal, Osmanabadi, Jakhrana, Jamunapari, Barbari, Marwari, Markhoz, Matongoats goat breeds, the 5-bp indel had low genetic diversities (0 < PIC < 0.164). In Boer, Haimen, Nubi, South group, North group, Foreign group as well as the backcrossed offspring, the PIC values ranged from 0.279 to 0.375 which illustrated that the 5-bp indel among these breeds had medium genetic diversities. Only in Sirohi goats, the 5-bp indel had high genetic diversity (PIC=1.133) [38,39]. Next, the association between different genotypes and growth traits was analyzed. The circumstance was that the 5-bp indel was distinctly related with body height, height at hip cross and chest depth index in lambs. It could be explained by that MSTN gene mutations gave rise to hyperplasia and hypertrophy which leaded to an increasing in muscle mass [51]. Additionally, we did a correlation analysis between growth traits which had a consistent phenotype with the body height and height at hip cross. It revealed that all growth traits had a positive correlation (P < 0.05) with each other. Hence, we speculate that the 5-bp indel can affect other growth traits of Shaanbei White Cashmere goats.”

[Question 26]

Line 149 please rephrase “in details” and again skip “goats” in each breed, instead, you can mention “goat breeds” at the end.

[Response 26]

Thank you for your advice. According to your suggestion, we rephrased the sentence:

“In details, in Black Bengal, Osmanabadi, Jakhrana, Jamunapari, Barbari, Marwari, Markhoz, Matongoats goat breeds, the 5-bp indel had low genetic diversities (0 < PIC < 0.164). In Boer, Haimen, Nubi, South group, North group, Foreign group as well as the backcrossed offspring, the PIC values ranged from 0.279 to 0.375 which illustrated that the 5-bp indel among these breeds had medium genetic diversities. Only in Sirohi goats, the 5-bp indel had high genetic diversity (PIC=1.133)”.

[Question 27]

Table 2 Can the authors explain why is it that the sample number for all phenotypes is 548 in the homozygotes, but just 73 on body weight? Bodyweight is a relevant trait for the MSTN gene, and… if the phenotypes registered for this trait where just 73 it should be stated somewhere in material and methods. Same observation in Appendix 2

[Response 27]

Thanks a lot for your careful review. Some body weight records provided by the farm were missing. Maybe the staff forgot to record. However, the sample size of the rest body weight records has met the statistical rationality (n > 30). So, we did an association analysis between the 5-bp indel and the body weight. We have stated this in “Material and Methods” and “Table S3”.

[Question 28]

Appendix 1 is impossible to read as it is, the numbers are not in the same line. Please accommodate the columns properly so the table can be read.

[Response 28]

Thank you very much for this suggestion. The columns of the Table S1 were carefully accommodated as you suggested.

[Question 29]

Line198: “being indel markers extensively used”

[Response 29]

Thanks a lot. The grammar mistake has been revised.

[Question 30]

In general, in the whole manuscript: avoid tendentious words that may establish decisive judgments, such as: absolutely (line 50), definitely (l.52), dramatically (l.200), remarkably (l.239) etc.

[Response 30]

Greatly appreciated with your review. Tendentious words in this article were all deleted.

[Question 31]

Line 200 is a major gene which/that regulate

[Response 31]

Thanks for reminding us of the grammar mistake. We have already modified it.

[Question 32]

In the results part, the authors do not give the results of the statistical model applied, they refer to appendix 3 for the correlation matrix but previously I miss results of one of the main analyses of the manuscript.

[Response 32]

Grateful for your opinion. The result of this Table S3 was added into the “Results”.

“Besides, we did a correlation analysis between growth traits of which “DD” genotype had a superior performance than “ID” genotype (BH, HHC, CC, BW). Results manifested that analyzed growth traits were positively correlated with each other (Table S3).”

[Question 33]

Appendix 3. Youth and Cannon bone circumference

[Response 33]

Greatly appreciated with your advice. We have changed the “Cannon circumference” to the “Cannon bone circumference”.

[Question 34]

From lines 198 to 208 the authors give a mixture that goes from an introduction and conclusions in between. I would consider removing this, as it is repetitive information or discussing previous studies regarding the results obtained in this study.

[Response 34]

Deeply indebted with your proposal. The mixture goes from an introduction and conclusions in between was removed.

With best regards!

Sincerely Yours,

Yi Bi (biyi0312@163.com),

Xianyong Lan (lanxianyong79@126.com),

Xiaoyue Song (songxiaoyue@yulinu.edu.cn) (corresponding author),

Chuanying Pan (panyu1980@126.com), et al.

College of Animal Science and Technology,

Northwest A&F University, Yangling, Shaanxi 712100, China

Reviewer 4 Report

The manuscript deals with an interesting topic as it seeks the association of Myostatin gene with traits of high economical interest. Furthermore, we must highlight the large sample used for the study, which is one of its strong points. Hence, I see this manuscript offers something that is worth considering for publication.

However, there are some ideas that must be considered before the paper could be ready to be published. I consider the introduction is well written and easily followed. However, I feel as if much of the weight had been deposited on the fact that a large sample is being use, rather than the repercussion of this analysis on economical traits. The first thing that I miss is the lack of a clearly developed statement for the objective in the abstract and at the end of the introduction (which is present but very succinct). It can be inferred, but you need to state what you did in study that you carried out and what is the most likely outcome.

Genotyping procedures are described in detail.

Still I have some concerns on the fact that these data, even if they have been obtained out of a large sample considering the studies that are found in literature, have not been tested for parametric assumptions (such as normality or homoscedasticity) and a parametric approach has been carried out directly, without thinking of how the data should be distributed.

Furthermore, the number of traits included is considerable, still no correction such as Bonferroni correction or methods such as PCA have been performed to identify potential redundancies in these factors to prevent for the occurring of Type I errors. For instance, indexes (such as those for cannon circumferences, chest width and hip width) may be closely related to the original measures themselves.

Hence, without these evidences we cannot conclude whether the conclusions drawn are valid or not.

As the paper presents an important topic, is well written, and presents a good design, I would suggest that the authors provide further information in regards the comments suggested, after which I could issue a rather solid verdict on the discussion and conclusions of this study.

Author Response

A cover letter with responses to the reviewers' comments and editorial comments on Animals-665915

Dear Editor and anonymous reviewers,

We greatly appreciate the anonymous reviewers for your careful review and constructive comments of our manuscript (ID: animals-665915). We have studied comments carefully and tried our best to revise this manuscript, and we hope that the revision can meet with your approval.

Here, we have listed the point-by-point responses to your detailed comments and suggestions (with red). As follows:

[Question 1]

The manuscript deals with an interesting topic as it seeks the association of Myostatin gene with traits of high economical interest. Furthermore, we must highlight the large sample used for the study, which is one of its strong points. Hence, I see this manuscript offers something that is worth considering for publication.

[Response 1]

Thank you very much for your valuable advice. We are pleased for your affirmation about the large sample sizes and more details about the large sample sizes has been added in the article.

[Question 2]

However, there are some ideas that must be considered before the paper could be ready to be published. I consider the introduction is well written and easily followed. However, I feel as if much of the weight had been deposited on the fact that a large sample is being use, rather than the repercussion of this analysis on economical traits. The first thing that I miss is the lack of a clearly developed statement for the objective in the abstract and at the end of the introduction (which is present but very succinct). It can be inferred, but you need to state what you did in study that you carried out and what is the most likely outcome.

[Response 2]

Feeling indebted for your opinion.

The abstract and introduction have been enriched as follows:

Abstract:

“Myostatin (MSTN) gene, also known as growth differentiation factor 8 (GDF8), is a member of the transforming growth factor-beta super-family and plays a negative role in muscle development. It acts at key points during pre-and post-natal life of amniotes that ultimately determine the overall muscle mass of animals. There are several studies concentrate on the effect of a 5-bp insertion/deletion (indel) within the 5’ untranslated region (5’UTR) of goat MSTN gene in goats. However, almost all the sample sizes were below 150 individuals. Only in Boer goats, the sample sizes reached 482. Hence, whether the 5-bp indel was still associated with the growth traits of goats in large sample sizes which were more reliable is not clear. To find an effective and dependable DNA marker for goat rearing, we first enlarged the sample sizes (n=1074, Shaanbei White Cashmere goat) which would enhance the robustness of the analysis and did the association analyze between the 5-bp indel and the growth traits. Results uncovered that the 5-bp indel was significantly related to the body height,height at hip cross and chest width index (P < 0.05). In addition, individuals with DD genotype had a superior growing performance than those with ID genotype. These findings suggested that the 5-bp indel in MSTN gene are significantly associated with growth traits and the specific genotype might be promising for maker-assisted selection (MAS) of goats.”

The End of Introduction:

“As the main genetic source of cashmere in local goat industry of North China, the Shaanbei White Cashmere goat which is a hybrid goat between Liaoning Cashmere goat (♂) and Shaanbei Ziwuling Black goat (♀), has the resistance to the crude feed, cold, wind-blown sand and disease [41]. Additionally, it has superior qualities and quantities of cashmere than other cashmere goat breeds. However, growth traits of Shaanbei White Cashmere goats need improving. Rapid, accurate as well as an effective method as MAS is. [42,43,44]. Herein, this study aimed at finding an effective and faithful DNA maker which could accelerate the process of selective breeding in goat rearing. A total of 1074 Shaanbei White Cashmere goats were analyzed, aiming at enhancing the robustness as well as revealing the impact of the 5-bp indel on the growth performance. These results may provide references for further research on applying MAS to the goat industry.”

[Question 3]

Genotyping procedures are described in detail.

[Response 3]

Thanks a lot. Specific modification was as follow:

“2.2. Primer design, PCR amplification and genotyping

Using the Primer Premier 6.0 (Premier, Canada), the primer pair (F: 5’- ACTGGTGTGGCAAGTTGTCTCT-3’; R: 5’-TTCCTTCTGCTCGCTGTTCTCA-3’) for amplification of indel loci in 5’UTR was designed based on the goat MSTN gene sequence (NC_030809.1) and the Ensemble Indel-database (http://apr2019.archive.ensembl.org/index.html/indel). Assays were performed by touch-down PCR in a 13 μL volume, containing 6.5 μL 2 × mix, 0.2 μL each of forward and reverse primers, 0.5 μL genomic DNA (20 ng/μL), and 5.6 μL ddH2O. The PCR protocol was as below: initial denaturation for 5min at 95 oC; followed by 18 cycles of denaturation for 30 s at 95 oC, annealing for 30 s at 68 oC (with a decrease of 1 oC per cycle), extension for 15s at 72 oC; another 38 cycles of 30s at 95 oC, 30s at 50 oC, and 15s at 72 oC and a final extension for 5 min at 72 oC, with subsequent cooling to 12 oC. Using 3.5% agarose gel electrophoresis, spoting 0.5μL and running for 1.5h, the 5-bp indel of goat MSTN gene was examined via electrophoresis at 80V voltage. Then genotypes were identified combining with sequencing.”

[Question 4]

Still I have some concerns on the fact that these data, even if they have been obtained out of a large sample considering the studies that are found in literature, have not been tested for parametric assumptions (such as normality or homoscedasticity) and a parametric approach has been carried out directly, without thinking of how the data should be distributed.

[Response 4]

Greatly appreciated with your pertinent and careful suggestion.

To make it more precise about how the data was applied, we established a linear model to determine whether the data fitted a nomal distribution which was the precondition of the association analyze in SPSS. The linear model was as follow:

“Yjkl = μ+ Gk + Ejkl, where Yjkl was the trait measured on each of the jklth animal; μ the overall mean; Gk the type of the kth gentotype, and Ejkl was the random error”.

[Question 5]

Furthermore, the number of traits included is considerable, still no correction such as Bonferroni correction or methods such as PCA have been performed to identify potential redundancies in these factors to prevent for the occurring of Type I errors. For instance, indexes (such as those for cannon circumferences, chest width and hip width) may be closely related to the original measures themselves.

Hence, without these evidences we cannot conclude whether the conclusions drawn are valid or not.

[Response 5]

Deeply Grateful for your advice.

Considering your suggestion carefully, we would like to discuss our opinion with you:

Given that the Bonferroni t-test was a test based on LSD. By adjusting the significance of each test, the total type I error probability is controlled at a certain level. But the fact that only two genotypes were found in this goat population. We chose to use independent sample t-test which was not include in the ANOVA. So, the multiple comparison was also ignored.

Any further suggestion was looking forward to bring forward.

Yours sincerely.

[Question 6]

As the paper presents an important topic, is well written, and presents a good design, I would suggest that the authors provide further information in regards the comments suggested, after which I could issue a rather solid verdict on the discussion and conclusions of this study.

[Response 6]

Thanks for so much affirmative and pertinent suggestion you gave for our manuscript. We accept the all proposal sincerely and look forwarding more discussion with you to improve the quality of the article.

Thanks a million.

With best regards!

Sincerely Yours,

Yi Bi (biyi0312@163.com),

Xianyong Lan (lanxianyong79@126.com),

Xiaoyue Song (songxiaoyue@yulinu.edu.cn) (corresponding author),

Chuanying Pan (panyu1980@126.com), et al.

College of Animal Science and Technology,

Northwest A&F University, Yangling, Shaanxi 712100, China

Round 2

Reviewer 2 Report

The authore have addressed the majority of my concerns. However, I still suggest the use of the HGVS nomenclature (http://varnomen.hgvs.org/) and a coding DNA reference sequence when descripting variation or SNPs in a gene.

For example, the nucleotide 5' of the ATG-translation initiation codon is -1, the previous -2, etc. The nucleotide 3' of the translation stop codon is *1, the next *2, etc.

Beginning of the intron; the number of the last nucleotide of the preceding exon, a plus sign and the position in the intron, like c.77+1G, c.77+2T, etc.

End of the intron; the number of the first nucleotide of the following exon, a minus sign and the position upstream in the intron, like c.78-1G.

In the middle of the intron, numbering changes from "c.77+.." to "c.78-.."; for introns with an uneven number of nucleotides the central nucleotide is the last described with a "+".

Author Response

A cover letter with responses to the reviewers' comments and editorial comments on Animals-665915-R2

Dear Editor and anonymous reviewers,

We greatly appreciate the anonymous reviewers for your careful review and constructive comments of our manuscript (ID: animals-665915). We have studied comments carefully and tried our best to revise this manuscript, and we hope that the revision can meet with your approval.

Here, we have listed the point-by-point responses to your detailed comments and suggestions (with red). As follows:

[Question 1]

The authore have addressed the majority of my concerns. However, I still suggest the use of the HGVS nomenclature (http://varnomen.hgvs.org/) and a coding DNA reference sequence when descripting variation or SNPs in a gene.

For example, the nucleotide 5' of the ATG-translation initiation codon is -1, the previous -2, etc. The nucleotide 3' of the translation stop codon is *1, the next *2, etc.

Beginning of the intron; the number of the last nucleotide of the preceding exon, a plus sign and the position in the intron, like c.77+1G, c.77+2T, etc.

End of the intron; the number of the first nucleotide of the following exon, a minus sign and the position upstream in the intron, like c.78-1G.

In the middle of the intron, numbering changes from "c.77+.." to "c.78-.."; for introns with an uneven number of nucleotides the central nucleotide is the last described with a "+".

[Response 1]

Greatly appreciated with your recommendation. We have already modified the description of the 5-bp indel as you suggested. Specific modification are as follows:

“Therefore, mutations of MSTN gene will always be one of the hot spots. A 5-bp indel (c.-120ins) in the 5’ Untranslated region (5’UTR) of goat MSTN gene was reported to related to the growth traits of goat.”

“A 5-bp indel (c.-120ins) in the 5’ UTR of goat MSTN gene was detected. There were two different genotypes: homozygote deletion type (deletion/deletion: DD, 206 bp); heterozygote type (insertion/deletion: ID, 211 bp and 206 bp) (Figure.1).”

We sincerely hope that the current revised version can be considered to be acceptable for publication in Animals.

With best regards!

Sincerely Yours,

Yi Bi (biyi0312@163.com),

Xianyong Lan (lanxianyong79@126.com),

Xiaoyue Song (songxiaoyue@yulinu.edu.cn) (corresponding author),

Chuanying Pan (panyu1980@126.com), et al.

College of Animal Science and Technology,

Northwest A&F University, Yangling, Shaanxi 712100, China

Reviewer 3 Report

Changes were made according my suggestions, although regarding statistical significance on my comments, just redaction was changed on the paper, rather than proper statistical analyses.

Author Response

A cover letter with responses to the reviewers' comments and editorial comments on Animals-665915

Dear Editor and anonymous reviewers,

We greatly appreciate the anonymous reviewers for your careful review and constructive comments of our manuscript (ID: animals-665915). We have studied comments carefully and tried our best to revise this manuscript, and we hope that the revision can meet with your approval.

Here, we have listed the point-by-point responses to your detailed comments and suggestions (with red). As follows:

[Question 1]

Changes were made according my suggestions, although regarding statistical significance on my comments, just redaction was changed on the paper, rather than proper statistical analyses.

[Response 1]

Thank you very much for your meticulous suggestion.

Combing your pertinent and careful suggestion and other reviews’ recommendation, we finally chose to used independent-samples T test to do the association analyze. Furthermore, we made sure all the records were in the normal distribution which met the precondition to do the association analyze. In details:

we established a linear model:

“Yjkl = μ+ Gk + Ejkl, where Yjkl was the trait measured on each of the jklth animal; μ the overall mean; Gk the type of the kth gentotype, and Ejkl was the random error”

and then did a normality test. After identifying all the data were in the normal distribution, we used the independent-samples T test to do an association analyze which was recommended by other reviewers.

The specific results of the normality test were as follow:

For adults:

For kids:

Sincerely hope the current revised version can be considered to be acceptable for publication in Animals.

With best regards!

Sincerely Yours,

Yi Bi (biyi0312@163.com),

Xianyong Lan (lanxianyong79@126.com),

Xiaoyue Song (songxiaoyue@yulinu.edu.cn) (corresponding author),

Chuanying Pan (panyu1980@126.com), et al.

College of Animal Science and Technology,

Northwest A&F University, Yangling, Shaanxi 712100, China

Reviewer 4 Report

It is true that The Bonferroni Test is a multi-comparison test. It allows you to compare, like other contrasts of this type, the means of the t levels of a factor after having rejected the null hypothesis of average equality using the ANOVA technique.

All multiple comparison tests are tests that try to concrete a generic alternative hypothesis like that of any of the ANOVA Tests.

However, Fisher's LSD test is not based on The Bonferroni Test, but must however, be understood in relation to it, as is based on the creation of a threshold, the BSD (Bonferroni significant difference) above which, like the LSD in the LSD Test, the difference between the two means will be significant and below which that difference will not be statistically significant.

If you compare both tests you will see that the differences occur at the chosen significance level. In the Bonferroni Test the significance level is modified based on the number of comparisons to make. This eliminates the problem of making multiple comparisons. Reduces the level of significance to such an extent that it eliminates the error of applying the test so many times at the same time.

The peculiarity of LSD technique is the reduction of the significance level, the division of the usual alpha level by M, the total number of possible comparisons from two to two. This compensates for the possible error that can be made by making many comparisons two to two, each with that predetermined possibility of alpha error.

Once this has been clarified, the problem has not been addressed yet.

Let’s say you suggested that you established a linear model to determine whether the data fitted a nomal distribution which was the precondition of the association analyze in SPSS.

The word Whether implies that your model either can or cannot fulfill the assumption, but then anything is said in regards what is the real distribution (normal or not) of the data for this study.

Furthermore, another of the important assumptions to test is homoscedastidity. It is true that fulfilling one of both (normality or homoscedastidity) when the rest of assumptions have been fulfilled (lack of outliers, no multicollinearity, among others) could be considered enough. However, when both fail, ANOVA, indiependent t-tests or any other parametric test are not an option, unless you tranform the data and backtransform it to report results and draw conclusions.

If you have a factor with two levels, as it happens in this case, then the most interesting option to apply is Mann-Whitney test (when normality and homoscedastidity have been violated).

Again Please clarifiy if data was normally distributed and homoscedastic. Otherwise, the statistical analysis must be changed. Otherwise, results may change and conclusions may not be valid.

Author Response

A cover letter with responses to the reviewers' comments and editorial comments on Animals-665915

Dear Editor and anonymous reviewers,

We greatly appreciate the anonymous reviewers for your careful review and constructive comments of our manuscript (ID: animals-665915). We have studied comments carefully and tried our best to revise this manuscript, and we hope that the revision can meet with your approval.

Here, we have listed the point-by-point responses to your detailed comments and suggestions (with red). As follows:

[Question 1]

It is true that The Bonferroni Test is a multi-comparison test. It allows you to compare, like other contrasts of this type, the means of the t levels of a factor after having rejected the null hypothesis of average equality using the ANOVA technique.

All multiple comparison tests are tests that try to concrete a generic alternative hypothesis like that of any of the ANOVA Tests.

However, Fisher's LSD test is not based on The Bonferroni Test, but must however, be understood in relation to it, as is based on the creation of a threshold, the BSD (Bonferroni significant difference) above which, like the LSD in the LSD Test, the difference between the two means will be significant and below which that difference will not be statistically significant.

If you compare both tests you will see that the differences occur at the chosen significance level. In the Bonferroni Test the significance level is modified based on the number of comparisons to make. This eliminates the problem of making multiple comparisons. Reduces the level of significance to such an extent that it eliminates the error of applying the test so many times at the same time.

The peculiarity of LSD technique is the reduction of the significance level, the division of the usual alpha level by M, the total number of possible comparisons from two to two. This compensates for the possible error that can be made by making many comparisons two to two, each with that predetermined possibility of alpha error.

Once this has been clarified, the problem has not been addressed yet.

Let’s say you suggested that you established a linear model to determine whether the data fitted a nomal distribution which was the precondition of the association analyze in SPSS.

The word Whether implies that your model either can or cannot fulfill the assumption, but then anything is said in regards what is the real distribution (normal or not) of the data for this study.

Furthermore, another of the important assumptions to test is homoscedastidity. It is true that fulfilling one of both (normality or homoscedastidity) when the rest of assumptions have been fulfilled (lack of outliers, no multicollinearity, among others) could be considered enough. However, when both fail, ANOVA, indiependent t-tests or any other parametric test are not an option, unless you tranform the data and backtransform it to report results and draw conclusions.

If you have a factor with two levels, as it happens in this case, then the most interesting option to apply is Mann-Whitney test (when normality and homoscedastidity have been violated).

Again please clarifiy if data was normally distributed and homoscedastic. Otherwise, the statistical analysis must be changed. Otherwise, results may change and conclusions may not be valid.

[Response 1]

Deeply indebted to your pertinent suggestion.

Combing with your pertinent and careful suggestion and other reviews’ recommendation, we finally chose to used independent-samples T test to do the association analyze. Furthermore, we made sure all the records were in the normal distribution. In details:

we established a linear model:

“Yjkl = μ+ Gk + Ejkl, where Yjkl was the trait measured on each of the jklth animal; μ the overall mean; Gk the type of the kth gentotype, and Ejkl was the random error”

and then did a normality test. After identifying all the data were in the normal distribution, we used the independent-samples T test to do an association analyze which was recommended by all reviewers.

The specific results of the normality test were as follow:

For adults:

For kids:

Thank you very much for all the pertinent suggestion you put forward for the manuscript and sincerely hope the current revised version can be considered to be acceptable for publication in Animals.

With best regards!

Sincerely Yours,

Yi Bi (biyi0312@163.com),

Xianyong Lan (lanxianyong79@126.com),

Xiaoyue Song (songxiaoyue@yulinu.edu.cn) (corresponding author),

Chuanying Pan (panyu1980@126.com), et al.

College of Animal Science and Technology,

Northwest A&F University, Yangling, Shaanxi 712100, China
